# Sustainable Alternative Media for the Production of Lipolytic Cells and Fatty Acid Concentrates: Integration of the Enzyme and Food Industries

**DOI:** 10.3390/foods14060990

**Published:** 2025-03-14

**Authors:** Willian S. M. Reis, Arthur O. Preto, Giovanna M. Sant’Ana, Ikaro Tessaro, Ana L. G. Ferreira, Ernandes B. Pereira, Ana K. F. Carvalho

**Affiliations:** 1Department of Biotechnology, Lorena School of Engineering—USP, Lorena 12602-810, SP, Brazil; 2Faculty of Pharmaceutical Sciences, Federal University of Alfenas, Alfenas 37130-001, MG, Brazil; arthur.preto@sou.unifal-mg.edu.br (A.O.P.); ernandes.pereira@unifal-mg.edu.br (E.B.P.); 3Department of Chemical Engineering, Lorena School of Engineering—USP, Lorena 12602-810, SP, Brazil; giovannasantana@usp.br (G.M.S.); ikaro.tessaro@usp.br (I.T.); 4Department of Basic and Environmental Sciences, Lorena School of Engineering—USP, Lorena 12602-810, SP, Brazil

**Keywords:** lipase, agro-industrial by-products, hydrolysis, ultrasound

## Abstract

The use of agro-industrial by-products and processing residues, which are rich in carbohydrates, proteins, and lipids, in the production of lipases allows the sustainable use of these residues, reducing environmental impacts. In this study, the immersion water of lentils, soybeans, and textured soy protein was evaluated as carbon and nitrogen sources in the production of whole-cell lipases, and the resulting biomass was used in the hydrolysis of residual soybean oil with conventional heating and ultrasound. The results showed that the best culture medium was the one with 50% textured soybean protein, reaching values of 149.04 U/g of hydrolytic activity, 12.92 g/L of biomass concentration, 144.17 U of total biomass activity, and specific and volumetric productivities of 2.07 U/g·h and 20.02 U/L·h, respectively. The positive effect of adding soybean frying oil to the crop was observed, which increased cell production and hydrolytic activity. The biomass obtained showed potential for the ultrasound-assisted hydrolysis of vegetable oils, reaching approximately 43.36% hydrolysis in 7 h of reaction, with an initial rate of 31.03 mmol/h. It is concluded that soybean protein processing water is a viable candidate to replace traditional nitrogen sources, being an economically attractive alternative due to its wide generation in restaurants.

## 1. Introduction

The application of enzymes as biocatalysts in industrial processes is one of the main fields of interest in industrial biotechnology due to the search for alternatives to processes that use chemical catalysts that may require high pressures and temperatures. Characteristics such as high specificity and selectivity, low toxicity, product purity, reduced environmental impacts, and operation in moderate pH and temperature ranges make enzymes promising catalysts for a wide variety of applications [1,2].

Among the enzymes used in industrial processes, lipases stand out, with a global market that is constantly growing. In 2024, the market was estimated to be worth almost USD 690 million, and projections indicate that it should reach USD 1397.2 million, with an annual growth of 12.5% [3]. Lipases (EC 3.1.1.3) are enzymes of the hydrolase class capable of hydrolyzing triacylglycerol ester bonds into free fatty acids (FFAs) and glycerol in aqueous media and under water-restricted conditions, and can catalyze reactions such as esterification, transesterification, aminolysis, and glycerolysis. With excellent stability and activity in the presence of organic solvents, lipases act on a wide range of substrates, including triacylglycerides, fatty acid esters, and natural or synthetic oils [2,4,5].

One of the main applications of these enzymes is in the hydrolysis of vegetable oils, whose products—free fatty acids and glycerol—are important precursors in the pharmaceutical, cosmetic, and oleochemical industries, and in the production of biodiesel [6], biosurfactants [7], aromatic esters [8], lubricants [9], and structured lipids [10].

Lipases can be of animal, microbial, or plant origin, with varying properties depending on their source [11]. Lipases, especially microbial lipases, have gained much more attention industrially than those derived from plants and animals due to their desirable characteristics and functional capacity under extreme conditions (pH and temperature), stability in organic solvents, chemoselectivity, enantioselectivity, and no requirement for any cofactor [12,13]. Microbial lipases can be produced in a solid state or submerged fermentation, with submerged fermentation having the advantages of high productivity, easy operability, and better process control [14].

Fungi are potential producers of lipases with unique catalytic properties that are important for commercial applications. Most commercially important lipase-producing fungi belong to genera such as *Rhizopus*, *Aspergillus*, *Penicillium*, *Geotrichum,* and *Mucor*. Filamentous fungi, in particular, are good lipase producers, and extraction, purification, and processing can be relatively simple [12]. The lipases produced can be extracellular or intracellular lipases, and lipases in their free form often present operational instabilities (pH and temperature), in addition to not allowing the reuse of the biocatalyst that is soluble in the reaction medium and may require immobilization steps to make their application viable [12,15]. Intracellular lipases include mycelium-bound lipases, defined as lipases that are associated with the fungal biomass and are therefore naturally immobilized. Although bound to the mycelium, the enzyme is still active, and so it can be used as a biocatalyst, partially eliminating the purification, recovery, and immobilization steps [16,17,18].

The culture media employed for lipase synthesis through submerged fermentation typically require the inclusion of a carbon source, a nitrogen source, and a lipid source as essential components for optimal enzyme production. Although synthetic media, such as glucose and soy peptone, are commonly used, there is growing interest in the reutilization of agro-industrial waste as nutrients for microbial growth and lipase induction. This approach is particularly appealing in emerging countries, where significant amounts of such waste are generated daily without proper treatment [19].

Studies show that kitchen wastewater and agricultural waste, such as orange peels and peanuts, can be used as substrates for the production of lipases by different microbial species [19,20]. Fibriana et al. [19] used wastewater from home kitchens, restaurants, and food factories as lipase inducers for the submerged cultivation of different microbial species, while Akinduyite et al. [20] used agricultural waste (orange peels, spoiled mangoes, and peanut shells) as a carbon source for the submerged fermentation of *Pichia kudriavzevii*, *Candida orthopsilosis* and *Saccharomyces cerevisiae*.

The growing demand for plant-based products, such as textured proteins, vegetable milks, and oils, reflects the behavior of a portion of the population that is looking for healthier and more sustainable alternatives. Innovations in soy processing, such as fermentation and enzymatic treatment, have enabled the development of functional and symbiotic foods, increasing their health benefits and commercial appeal [21,22]. The growth in the consumption of soy-derived products has transformed the food industry. Alternatives such as vegetable oils, textured proteins, and vegetable milks are gaining popularity among consumers looking for healthier and more sustainable options [23,24].

Brazil, the world’s largest producer and exporter of soybeans, with more than 150 million tons per year, is an example of the integration of sustainability and the circular bioeconomy. In addition to oil, soy protein, bran, and by-products such as defatted bran and soy hulls are used in sustainable practices, reducing waste and maximizing the use of the plant [25,26]. Agro-industrial soybean waste includes hulls, cake, press cake, and industrial effluents, which often contain oils and fats [27].

The environmental impact of soybean production is significant, given the high water consumption in processing and the increase in by-products such as soybean cake and industrial effluents. Sustainable practices, such as the reuse of waste, have promoted a circular economy, mitigating environmental impacts [28,29,30]. Thus, by-products (liquid and solid), effluents, and wastewater from soybean processing, due to their high contents of carbohydrates, proteins, and lipids, can be strategically used as substrates in cultivation media for the production of enzymes, especially lipases [29,31].

The aim of this study was to produce whole cells of the fungus *Mucor circinelloides* with lipolytic activity through submerged cultivation using waste products from the soybean production biorefinery as sources of carbon and nitrogen. The resulting biomass was used in hydrolysis reactions of residual soybean oil using conventional and ultrasound heating.

## 2. Materials and Methods

### 2.1. Microorganism

The strain used was the fungus *Mucor circinelloides* URM4140 obtained from the mycotheque of the Federal University of Pernambuco (URM-UFPE) ( Brazil). In order to obtain and maintain culture spores, fungal cells had been previously inoculated on Sabouraud agar medium under aseptic conditions. The culture was incubated at 30 °C and 72 h, or until it reached the highest sporulation status. Cells were washed with 10 mL of sterile distilled water to obtain spore suspension under aseptic conditions.

### 2.2. Materials

Olive oil (Carbonell™), soybeans, lentils, and textured soy protein were purchased at a local market (Brazil) and residual soybean oil was obtained from the university restaurant of the Lorena School of Engineering (Brazil), being a residual oil that originated from deep-frying. The residual soybean oil has the following fatty acid composition (wt): 11.5% palmitic acid, 4.1% stearic acid, 23.5% oleic acid, 53.3% linoleic acid, and 6.8% linolenic acid, with a 276 g mol/L average molecular weight. After collection, the residual oil was not subjected to any further treatment. Sabouraud agar medium and soybean peptone were acquired from HiMedia Laboratories ( India). Gum arabic, monobasic potassium phosphate, monobasic sodium phosphate, and bibasic sodium phosphate were acquired from Dinâmica Química (Brazil), and magnesium sulfate heptahydrate, sodium hydroxide, sodium nitrate, and an ethanol solution (70% *v*/*v*) were obtained from Vetec Química (Brazil). All other reagents and organic solvents of analytical grade were purchased from Vetec Química.

### 2.3. Enzymatic Activity

Mycelium-bound lipase activity was assessed in terms of its dry biomass concentration (g/L) and hydrolytic activity (U/g) using the method of olive oil emulsion hydrolysis. Enzyme activity (U/g) is defined as the amount of dry biomass required to release 1 μmol of free fatty acids per minute under experimental conditions (0.1 g of biomass at a 37 °C reaction temperature, 100 mM sodium phosphate buffer, and a pH of 7.0 for a 5 min reaction) [32].

### 2.4. Culture Medium and Experimental Conditions

#### 2.4.1. Culture Medium

Firstly, the soybeans, lentils, and textured soy protein (TSP) were soaked to extract the water-soluble compounds. For this, 100 g of soybeans and lentils were thoroughly washed under running water to remove any impurities. The beans were then placed in separate containers and covered with 500 mL of water. Both types of grains were left to soak overnight at room temperature, after which the soaking water was collected by vacuum filtration. The culture medium for TSP was prepared according to the manufacturer’s instructions by adding 360 mL of hot water to 120 g of soy protein. The protein was submerged for 15 min, and the soy water was obtained by filtering the textured protein granules. After the immersion period, the water was filtered and collected for further use. The soaking water was kept refrigerated at 4 °C until the moment of inoculation, ensuring the preservation of the nutrients, minerals, and bioactive compounds present. To prepare the culture, the immersion water was previously autoclaved at 121 °C for 15 min.

Different culture media were evaluated for the production of lipolytic cells using media with 100%, 50%, and 25% immersion water, and the media were diluted with sterile distilled water. In addition, a conventional medium was prepared to compare the lipase production efficiency of *M. circinelloides* in the alternative media. The conventional medium was composed of 30 g/L olive oil, 70 g/L soy peptone, 1 g/L NaNO_3_, 1 g/L KH_2_PO_4_, and 0.5 g/L MgSO_4_-7H_2_O, all autoclaved at 121 °C for 15 min [33].

#### 2.4.2. Experimental Conditions for Submerged Cultivation

*Mucor circinelloides* cultivations were performed in 250 mL Erlenmeyer flasks containing 100 mL of autoclaved medium (conventional media and alternative culture media) and inoculated with a suspension of 10^6^ spores at 30 °C and orbital shaking at 180 rpm for 72 h. The spore concentration was determined in a Neubauer chamber using an Olympus^®^ binocular microscope (Olympus Corporation, Tokyo, Japan) [33]. At the end of the fermentation process, the produced biomass was separated from the medium by vacuum filtration and the humidity was quantified by drying the wet biomass (0.25 g) in a microwave oven (180 W per 5 min) [34]. Subsequently, the fungal biomass was stored at 4 °C prior to use.

### 2.5. Lipase Production Assessment

Lipase production by *M. circinelloides* was evaluated in terms of the hydrolytic activity of the fermentation broth (extracellular lipase; U/g) and biomass (whole-cell hydrolytic activity; U/g) using the olive oil emulsion hydrolysis method. Enzymatic activity (U/g) is defined as the amount of dry biomass required to release 1 μmol of free fatty acids per minute under specific experimental conditions (0.1 g of biomass at 37 °C using 100 mM sodium phosphate buffer, pH 7.0, and a 5 min reaction) [32].

Furthermore, the biomass produced in the cultivation of *M. circinelloides* was analyzed for total biomass activity (TBA) (U), which represents the overall activity generated by the biomass (Equation (1)); specific enzyme productivity (SEP) (U/g·h), which indicates the rate of activity production per unit of biomass in a given time interval; and volumetric enzyme productivity (VEP) (U/L·h), which reflects the amount of enzyme produced per unit volume of culture medium in a specific period. Each of these parameters was calculated from Equations (1)–(3) described below.(1)$$Total\;Biomass\;Activity\left(U\right)=Activity\left(\frac{U}{g}\right)\times Dry\;Biomass\left(\frac{g}{L}\right)$$(2)$$Specific\;Enzyme\;Productivity\left(\frac{U}{g\cdot h}\right)=\frac{Activity\left(\frac{U}{g}\right)}{time\;(h)}$$(3)$$Volumetric\;Enzyme\;Productivity\left(\frac{U}{L\cdot h}\right)=\frac{Activity\left(\frac{U}{g}\right)\times Biomass\;Concentration\left(\frac{g}{L}\right)}{time\;(h)}$$

### 2.6. Evaluation of Residual Soybean Oil and Textured Soybean Protein in Lipase Production

In order to evaluate the viability of using two byproducts from soybean processing, residual soybean oil (30 g/L) was tested as a substitute for olive oil, while textured soy protein was used as a substitute for soy peptone in the traditional culture medium. The culture medium and experimental conditions were prepared as described in Section 2.3, Section 2.4.1 and Section 2.4.2. At the end of the cultivation, the biomass concentration, hydrolytic activity, enzymatic productivity, and total biomass activity were evaluated.

### 2.7. Hydrolysis of Residual Oil in Stirred Tank Reactors with Conventional Heating and Ultrasound Bath

The hydrolysis reactions were performed using conventional heating and ultrasound systems (Model USC 1800-A Ultrasonic Cleaner (Unique, Brazil) and whole cells of *M. circinelloides* as the biocatalyst obtained from the cultivation described in Section 2.5. In 250 mL glass-jacketed reactors, 100 mL of the substrate composed of an emulsion of 25 g of vegetable oil in 100 mM sodium phosphate buffer, pH 6.0, using gum arabic as the emulsifier (3% *m*/*v*) was prepared. The tests were performed at 40 °C with a fixed ratio of 200 U activity units, and 600 rpm of mechanical stirring was performed using an overhead motor stirrer with a steel helical impeller. A 50:50 (*v*/*v*) mixture of acetone and ethanol was added to aliquots (0.5 g) removed periodically, and the fatty acid content was quantified by titration with a 20 mM sodium hydroxide solution (NaOH) using phenolphthalein as indicator. Ultrasonic heating reactions were performed under the same conditions, such as temperature and time, with a power of 100 W. The percentage of hydrolysis (%), according to [35], was calculated by Equation (4):(4)$$Hydrolysis\left(\%\right)=\frac{\left(V_{a}-V_{b}\right).C_{NaOH}.10^{-3}\;.M}{m\;.f}$$
where V_a_ is the volume of NaOH in the sample (mL); V_b_ is the volume of NaOH in the control (mL); C_NaOH_ is the molar concentration of NaOH (25 mmol/L); M is the average molecular mass of fatty acids in the vegetable oil; m is the mass of the sample (0.5 g); and f is the oil fraction (0.25).

The initial reaction rates were analyzed based on the formation of free fatty acids (mmol/L) in the first 7 h of the reaction. The results were plotted using Origin Pro software version 5.0^®^ to obtain the linear equation for the initial rates of the hydrolysis reaction. The calculation of free fatty acids is described according to Equation (5):(5)$$FFA\left(mmol/L\right)=\frac{V_{a}-V_{b}.C_{NaOH}.10^{3}}{m}$$
where V_a_ is the volume of NaOH in the sample (mL); V_b_ is the volume of NaOH in the control (mL); C_NaOH_ is the molar concentration of NaOH (20 mmol/L); and m is the mass of the sample (0.5 g).

### 2.8. Statistical Analysis

Experiments were conducted in triplicate and the results were analyzed using the software Statistica^®^ version 14.1.0 with Tukey’s test.

## 3. Results

### 3.1. Evaluation of Alternative Culture Media as Sources of Carbon and Nitrogen

This study presents an initial assessment to investigate the feasibility of using wastewater from the washing and processing of legumes, such as lentils, soybeans, and soy protein, as alternative nutrient sources for lipase production. These waters, which contain organic compounds and nutrients derived from legume processing, may offer a sustainable approach that promotes the reuse of by-products and the reduction of production costs. The preliminary analysis aims to evaluate the potential of these water sources to support microbial growth and lipase production, while also assessing the impact of their composition on the efficiency of the biotechnological process.

The application of wastewater from the preparation of lentils, soybeans, and soy protein in the production of lipase linked to the mycelium of *M. circinelloides* was analyzed. The culture media were evaluated in different proportions (25%, 50%, and 100%, (*v*/*v*)) of wastewater, in addition to a conventional culture medium that was previously described in the literature as efficient in the production of lipolytic cells [33]. Biomass production, as well as extracellular and cellular biomass hydrolytic activities, were evaluated as parameters to determine the effectiveness of the different culture conditions. The results obtained are shown in Figure 1.

Figure 1A shows that the greatest production of hydrolytic activity was retained in the mycelium of the fungal biomass, since for all culture media evaluated, the extracellular activity was less than 13 U/g reported, with the greatest activity being obtained with the conventional medium. However, production in conventional medium and that consisting of 100% soybean wastewater did not show statistically significant differences (*p* > 0.05). This result was expected and is in line with the findings previously described in similar studies using defined culture media, as reported by different studies that evaluated several filamentous fungi. The low production of hydrolytic activity of extracellular lipases of some strains of filamentous fungi, such as *Rhizopus orzyae* [36,37], *Penicillium citrinum* [32], *Aspergillus flavus* [38], and *Mucor circinelloides* [37], has been described.

As mentioned, the greatest hydrolytic activity was retained by the mycelium of the fungal biomass, with the results showing a significant variation in the activity of the lipase bound to the mycelium among the different culture media and concentrations evaluated. Among the cultures studied, the conventional medium showed the greatest biomass activity of 173 U/g, followed by the cultivation in water of soy protein (50%) (149.04 U/g), lentils (100%) (135.82 U/g), and soy protein (100%) (134.77 U/g), with the last two media evaluated showing no statistically significant differences. These results suggest that higher concentrations of legume preparation water in the culture medium favor the production of lipase, which may be related to specific absorption and metabolic characteristics of these ingredients or to the way in which vegetable proteins interact with cells. However, crops with low concentrations of lentils and soybeans (25%) presented the lowest values of hydrolytic activity, which were significantly lower. These differences indicate that the culture medium with lentil water and soybean water in lower concentrations tend to generate lower values of hydrolytic activity of the lipase bound to the fungal mycelium, probably due to the low availability of nutrients.

The results obtained in this work are also similar to those reported in the literature with different filamentous fungi but using cultivation in conventional medium with commercial soybean peptone and refined vegetable oil. Castro et al. [39], in the cultivation of the fungus *Penicillium purpurogenum*, achieved a lipolytic activity of 150.70 U/g. Lima et al. [32] obtained activity values between 80.12 and 256.81 U/g for the fungus *Penicillium citrinum* and Braz et al. [40], with the cultivation of the fungus *M. circinelloides*, obtained 160 U/g of hydrolytic activity. Similar hydrolytic activities were also shown in studies involving the production of lipases from whole cells of *R. oryzae* but using defined media, where activity values of 120–140 U/g [41] and 100–122 U/g [18] were obtained.

Soybeans are mainly composed of carbohydrates (42.4% to 48.1%), proteins (34.2% to 35.4%), and lipids (13.1% to 17.5%) [42]. In comparison, lentils contain a higher proportion of carbohydrates (60%), but lower percentages of proteins (22%) and lipids (2.2%) [43]. Thus, soybeans stand out for their significant oil and protein contents, while lentils are rich in carbohydrates and fiber, with a lower amount of lipids. These data can also help in understanding the biomass concentration results shown in Figure 1B, where the soybean trials presented the highest biomass concentrations.

Among the liquid byproducts from soybean processing, only refined soybean oil has been used in the production of lipases from whole cells [33,44]. Some filamentous fungi that also produce mycelium-bound lipase have already shown satisfactory results using organic liquid substrates such as brown sugar, sugarcane juice, and sugarcane molasses [45]; sugarcane vinasse [46]; sugarcane molasses [47]; sugarcane bagasse hydrolysate [48]; and cheese whey [40].

Regarding biomass production (Figure 1B), the culture medium with lentil water showed a significantly lower average biomass, with the lowest value recorded for lentil water at 25%, which was only 5.09 g/L. This was notably lower than all other treatments, suggesting that lentil water may be less efficient in promoting biomass production. This could be attributed to its nutritional composition, which may not be as conducive to cell growth compared to soy proteins or conventional media. The trial using conventional medium again stands out, with an average biomass production of 9.82 g/L, which was higher than all other treatments, except for the trials with soybeans at higher concentrations. Soybeans at 50% and 100% concentrations presented similar values (12.92 and 13.17 g/L, respectively), indicating that higher soybean concentrations tend to yield more biomass. To identify the most suitable protocol for obtaining a biomass with high hydrolytic activity, the parameters of the specific productivity of cellular lipase (U/g·h), volumetric productivity (U/L·h), and total biomass activity (U) were analyzed. The results obtained are shown in Table 1.

In terms of specific cellular lipase productivity, it was observed that soybean (25%) presented the lowest rate of 1.29 U/g·h, while soy protein (50%) reached the highest rate of 2.07 U/g·h. This result suggests that soy protein may be more efficient in lipase production per unit of biomass. However, the comparison of volumetric productivity revealed that although soy protein (50%) showed satisfactory specific productivity, the volumetric productivity was lower than that presented by the conventional medium (23.59 U/L·h), which was the highest among all the tests.

Regarding the total biomass activity, notable differences were also observed. Soy protein (50%) showed a high total activity of 144.17 U, surpassing the other protocols, except the conventional condition, which reached the maximum value of 169.84 U. The high total activity observed for soy protein (50%) may indicate greater efficiency in the production of lipase-producing biomass.

Lentils showed an increase in productivity as the concentration increased, reaching 1.89 U/g·h (100%) in terms of specific productivity. This increase, however, was not observed in volumetric productivity and total activity, which were below those observed for soybeans and soy protein. The results for soybeans (50%) and soy protein (50%) remained more consistent in terms of volumetric productivity, while lentils performed slightly less well.

Comparing the parameters of productivity and total biomass activity between the different protocols adopted with protein sources, textured soy protein (50%) showed the best results. This superior performance may be related to the nutrient composition and the production process of textured soy protein, which is commercially prepared by the thermoplastic extrusion of flours, grains, and protein concentrates under conditions of heat and pressure [49]. During the immersion stage, this process may have favored a faster and more efficient release of nutrients, such as soluble proteins and carbohydrates, creating a more nutritious environment for the development of the microorganism and, consequently, increasing the production of lipases.

On the other hand, the cultures prepared with soybeans and lentils may have shown a more limited release of nutrients due to the complex structure of the cell wall of these grains [50]. The difficulty in absorbing water in the grains indicates that a longer immersion time or the application of a higher temperature during treatment would be necessary, which could increase the hydration index and promote cell breakdown. These adjustments would be essential to improve the release of nutrients and thus provide a higher concentration of the compounds essential for fermentation [51].

Textured soy protein has a nutritional composition that can be utilized by a fungal biomass for cell growth and enzyme synthesis. It contains (% m/m) proteins (50%) and fats (6 to 8%) as macronutrients, which are essential for cell development and the production of lipases and other enzymes. Additionally, micronutrients (% m/m) such as potassium (2.13%), phosphorus (0.679%), calcium (0.248%), magnesium (0.275%), and iron (0.0069%) play crucial roles in various metabolic functions, including enzyme activation and electron transport. Textured soy protein can also provide amino acids such as lysine, threonine, isoleucine, leucine, phenylalanine, and valine, which are fundamental for protein synthesis and the nutritional balance required for fungal growth and enzyme production.

Therefore, among the protocols evaluated, soy protein showed the best results in the production of lipolytic cells and was therefore selected to continue with the other stages of this study.

### 3.2. Application and Influence of Residual Soybean Oil on Lipase Production: Cultivation in Wastewater of Textured Soy Protein

Lipase production by filamentous fungi is significantly influenced by the choice of lipid substrates, such as vegetable oils, which supply fatty acids and other essential components that stimulate lipolytic activity. These substrates not only induce the secretion of mycelium-associated lipases but also promote fungal growth and enhance enzyme expression. The chemical composition of the oil, including factors such as acidity and triglyceride profile, plays a crucial role in the production and specificity of lipases, directly affecting their efficiency in various industrial applications [52]. After selecting soy protein (50%) as the main constituent of the culture medium for the fungus *M. circinelloides*, the use of residual soybean oil was evaluated in the production of lipases bound to the fungal mycelium. The preparation of the culture medium and the experimental conditions adopted are described in Section 2.4.1 and Section 2.4.2, respectively. The results obtained are shown in Figure 2.

As shown in Figure 2, increasing the concentration of soy protein in the culture medium had a positive effect only in the trials without the addition of residual soybean oil, with hydrolytic activity increasing from 78.89 to 163.52 U/g, representing an increase of more than 50%. The same effect could not be observed in the cultures in which residual oil was added, showing a 15% drop in relation to the activity shown in the cultures with a lower concentration of soy protein. The same behavior was observed for biomass production, in which the biomass produced without the presence of residual oil had a 2.5× higher production, reaching a concentration of 7.71 g/L; for the media with oil, similar to biomass activity, a decrease in the biomass concentration was observed.

This result can be explained by the fact that without the supplementation of oil in the culture medium, the microorganism only has the nutrients released by the immersion process, and so as the concentration of soy protein water increased, more nutrients became available to be assimilated by the fungus for the production of cells and lipase. In parallel with the addition of oil, a process of substrate inhibition may have occurred with the increase in the soy protein concentration and the addition of oil to the medium.

The addition of oil promoted an increase in activity for both soy protein concentrations evaluated. For soy protein (100%), an activity of 163.53 U/g was obtained, with the activity almost tripling in relation to the test with the lowest soy protein concentration (72.89 U/g). For the tests with 100% soy protein, the hydrolytic activity without oil was 111.47 U/g, while with oil, it rose to 138.18 U/g. Although the increase was more moderate than in the 50% formulation, the presence of oil resulted in a significant increase.

The results obtained in this study are consistent with those described in the literature on the addition and influence of the composition of vegetable oils on the production of lipase-producing whole cells [52]. Studies using oils with higher concentrations of unsaturated fatty acids, such as oleic and linoleic acids, can promote an increase in the production of cells with high lipase activity bound to the mycelium [53,54]. Therefore, refined oils such as olive oil (74.5% oleic acid and 9.8% linoleic acid) and canola oil (60.4% oleic acid and 21.2% linoleic acid) are commonly used as carbon sources and inducers in the production of whole cells of different filamentous fungi that produce lipases, such as *Mucor circinelloides* [37,44], *Rhizopus oryzae* [33,41], *Aspergillus westerdijkiae* [16], and *Penicillium sp*. [17,42]. In the study by Zeng et al. [55], *R. oryzae* cells grown in medium supplemented with oleic and linoleic acids showed higher methanolytic activity than that observed with cells whose medium was enriched with saturated fatty acids.

Residual soybean oil has higher concentrations of linoleic acid (46–58%) and oleic acid (19–26%) [56,57], which makes it an ideal candidate to be used as a carbon source and/or inducer of the production of lipolytic cells with high hydrolytic activity. Çağatay and Aksu [58] also evaluated the use of residual frying oil in the production of extracellular and mycelium-bound lipases of *Rhizopus arrhizus*, where lipase activity increased with the addition of residual oil, reaching a maximum of 520 U/g with 10 g/L of residual soybean oil, which represented a 30% increase compared to the absence of oil.

In order to select the most appropriate protocol to obtain a biomass with high hydrolytic activity, the parameters of specific productivity of cellular lipase (U/g·h), volumetric productivity (U/L·h), and total biomass activity (U) were analyzed. The results obtained are shown in Table 2.

Table 2 shows that the addition of residual oil resulted in significant increases in specific productivity (U/g·h), volumetric productivity (U/L·h), and total activity (U), indicating that the presence of oil promoted greater lipase production and, consequently, greater enzymatic activity. For the assay with 50% soybean protein, specific productivity increased from 1.01 U/g·h (without oil) to 2.27 U/g·h (with oil), and volumetric productivity increased from 3.02 to 23.24 U/L·h. Total activity also increased from 21.77 to 167.34 U. However, the increase in the soybean protein concentration in the assays without oil showed an increase in all parameters evaluated. This behavior was not observed in the assays with oil, where the addition of oil and the increase in soybean protein concentration caused a negative effect on all parameters. This may have occurred due to a substrate inhibition effect, which may have reduced biomass production and hindered substrate assimilation by cells.

The increases in productivity and total activity in the medium with 50% soy protein can be attributed to the more significant increases in biomass production and hydrolytic activity with the addition of oil. For some microorganisms, lipase production may be independent of the addition of lipid substrates, but the addition of oil can increase the level of lipolytic activity. Certain microorganisms may prefer lipid substrates (vegetable oils or fatty acids) rather than simple carbon sources, such as sugars, for lipase production [52,59].

The reuse of soybean protein wastewater as a nitrogen source in lipase production can significantly contribute to reductions in the volume and organic load of the effluent produced, which in turn lowers the costs associated with effluent treatment. This approach not only optimizes resource use but also provides an economically and environmentally responsible solution for the production of enzymes such as lipases. For instance, peptone (used as a conventional nitrogen source) can be considered a common organic pollutant found in wastewater and has traditionally been used as a conventional nitrogen source in fermentation processes. However, its discharge into the environment poses serious risks, as it increases the biological oxygen demand and nitrogen content in the water, contributing to eutrophication. Under anaerobic conditions, microorganisms can decompose peptone, producing unpleasant odors such as hydrogen sulfide, which negatively affects air quality [60].

### 3.3. Hydrolysis of Residual Oil with Conventional Heating and Using an Ultrasonic Bath in Stirred Tank Reactors

After evaluating the hydrolytic activity of the biomass produced with soy protein water (50%) and residual soybean oil, it was applied in hydrolysis reactions of the same residual soybean oil used in the culture medium. Two different heating systems were evaluated, one conventional and one heating system with an ultrasound bath. The results obtained are shown in Figure 3.

According to the results obtained, it can be seen that the use of the ultrasound system provided a higher hydrolytic yield when compared to the conventional heating system. In 12 h of reaction, the conventional heating system achieved only 31% hydrolysis, while the ultrasound system achieved approximately 43% hydrolysis in just 7 h of reaction. The formation of free fatty acids was also evaluated (Figure 3B), with the conventional system reaching 279.08 mmol/L of FFAs in 12 h of reaction and the ultrasound system reaching 398.93 mmol/L of FFAs in just 7 h of reaction. These results indicate that hydrolysis reactions of waste oils catalyzed by whole cells with lipolytic activity can be carried out in alternative heating systems such as the ultrasound system, achieving superior results to those obtained in conventional heating systems. The ultrasound system appears to reach a sort of saturation point in the conversion and formation of free fatty acids at around 7 h, suggesting that after this point, increasing the reaction time does not result in significant gains. The ultrasound system appears to reach a saturation point in the conversion and formation of free fatty acids at around 7 h, indicating that extending the reaction time beyond this does not produce significant additional gains. This characteristic highlights the efficiency of the ultrasound system, which achieves superior hydrolysis results, and consequently free fatty acids, in a shorter period of time, thus reducing the need for prolonged reaction times, as exhibited by the conventional system.

From the results of the FFAs formed, it was possible to calculate the initial reaction speeds for each heating system by linearizing the FFA data for each reaction system, with the results described in Table 3.

From the results in Table 3, it is evident that the initial reaction rates for the ultrasonic heating system were higher than those of the conventional system, with a rate of 70.26 mmol/h, which was more than double the 27.92 mmol/h rate observed with conventional heating. This represents an increase of over 150% in reaction speed when using ultrasound, highlighting its ability to intensify the reaction and promote more efficient hydrolysis in less time. These results can be attributed to the effect of ultrasonic waves, which provide greater stability in the emulsion during the hydrolysis reaction, preventing the separation of the substrate phases (water/oil). As a result, the interaction between the lipase bound to the biomass mycelium and the substrate is enhanced. Therefore, the use of ultrasound not only improves process efficiency but also increases the conversion rate, making it a more advantageous approach compared to conventional heating.

Similar results involving the hydrolysis of vegetable oils with whole cells in an ultrasound system have been reported in the literature. A study evaluated the hydrolysis of waste cooking oil using lipase produced by *Aspergillus niger* and ultrasound. Factors such as temperature, exposure time, and ultrasound power were optimized. Ultrasound treatment increased the hydrolytic activity of lipase by 320% using 50% power at 45 °C for 25 min. The best condition for hydrolysis resulted in 62.67 μmol/mL of free fatty acids in 12 h [61].

The application of ultrasound waves has been studied in biocatalytic processes, including those involving enzymatic catalysis, such as in the production of biodiesel [62], free fatty acids [63], and esters [64]. Ultrasound technology uses low-frequency sound waves (10–100 kHz) to generate pressure waves that create cavitation bubbles in the reaction mixture. The collapse of these bubbles near the interface of two immiscible fluids results in improved mass transfer, and so the effects of ultrasound play an important role in improving enzyme stability and catalytic activity [64,65].

The application of ultrasound in lipase-catalyzed synthesis not only increases the yield but also accelerates the reaction speed under specific conditions, being a technology applied to both free [66] and immobilized enzymes [67]. Although scarce, some studies involving enzymatic hydrolysis with whole cells have already reported the use of ultrasound, as Lima et al. [32] observed benefits when using ultrasonic irradiation during the hydrolysis of soybean oil, employing lipase attached to the mycelium of *P. citrinum*. While conventional treatment resulted in only 38% hydrolysis, the application of ultrasonic irradiation increased this figure to 96%, demonstrating a significant gain in process efficiency. De Castro et al. [39] studied the use of whole cells of *Penicillium purpurogenum* as a biocatalyst for the hydrolysis of vegetable oils with a high concentration of lauric acid under low-power ultrasound irradiation. Hydrolysis reached up to 90% in 7 h, providing high levels of fatty acids in a shorter time than those reported in the literature.

This preliminary study on enzymatic hydrolysis catalyzed by whole cells of *M. circinelloides* assisted by ultrasound yielded promising results. Ongoing studies are focused on evaluating the operational conditions during the hydrolysis process to further optimize its efficiency.

## 4. Conclusions

This study demonstrated the potential of using agro-industrial by-products, such as textured soy protein wastewater and residual soybean oil, as alternative substrates for the sustainable production of whole-cell lipases by filamentous fungi. The wastewater containing 50% textured soy protein in the culture medium yielded the best results for the fungal biomass, with a hydrolytic activity of 149.04 U/g, biomass concentration of 12.92 g/L, total biomass activity of 144.17 U, and specific and volumetric productivities of 2.07 U/g·h and 20.02 U/L·h, respectively. The addition of soybean frying oil to the culture showed a positive effect, increasing both cell production (from 3.08 to 7.71 g/L) and hydrolytic activity (from 78.89 to 163.52 U/g). The generated biomass demonstrated significant potential for the ultrasound-assisted hydrolysis of vegetable oils, achieving approximately 43.36% hydrolysis in 7 h of reaction, with an initial rate of 31.03 mmol/h, highlighting the potential of ultrasound as an alternative heat source in whole-cell catalyzed hydrolysis reactions. The integration of the food industry with lipase production from agro-industrial waste offers a sustainable solution, adding value to by-products and optimizing industrial processes.

## Figures and Tables

**Figure 1 foods-14-00990-f001:**
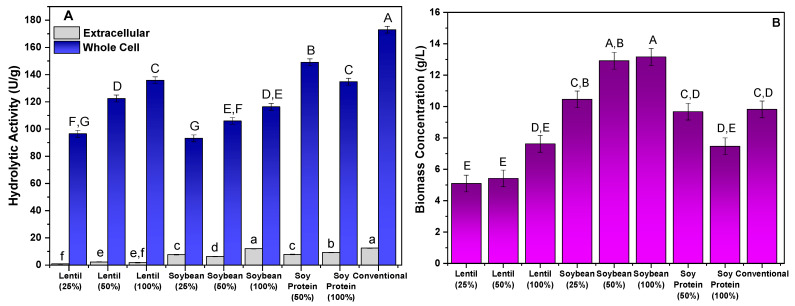
Evaluation of alternative culture media in the production of lipases bound to the mycelium of *M. circinelloides*: hydrolytic activity (U/g) (**A**) and biomass concentration (g/L) (**B**). Different letters indicate a significant difference (*p* < 0.05) between protocols.

**Figure 2 foods-14-00990-f002:**
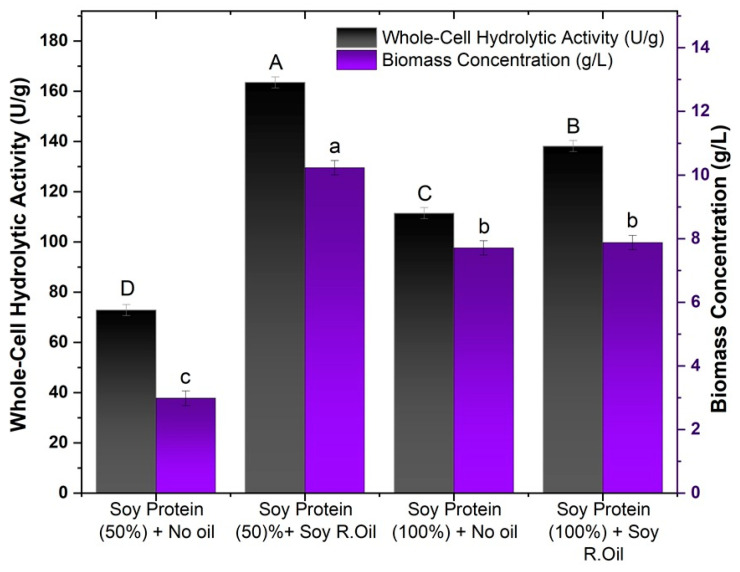
Effects of residual soybean oil and textured soy protein wastewater on biomass production and hydrolytic activity. Different letters indicate a significant difference (*p* < 0.05) between protocols.

**Figure 3 foods-14-00990-f003:**
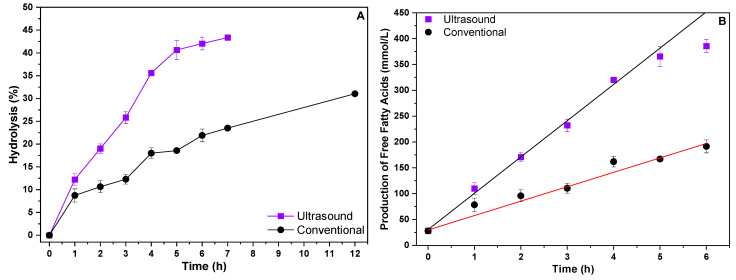
Enzymatic hydrolysis (**A**) of residual soybean oil and evaluation of fatty acid production (**B**) in different heating systems.

**Table 1 foods-14-00990-t001:** Comparison of specific lipase productivity, volumetric productivity, and total biomass activity using different protein sources.

Protein Source	Medium Concentration (%)	Specific Productivity (U/g·h)	Volumetric Productivity (U/L·h)	Total Biomass Activity (U)
Lentil	25%	1.34	6.83	50.45
	50%	1.70	9.21	79.09
	100%	1.89	14.38	89.41
Soybean	25%	1.29	13.53	23.73
	50%	1.47	19.00	28.69
	100%	1.62	21.27	44.35
Soy Protein	50%	2.07	20.02	144.17
	100%	1.87	13.97	100.58
Conventional	***	2.40	23.59	169.84

*** there is no data.

**Table 2 foods-14-00990-t002:** Effect of adding residual soybean oil: comparison of extracellular activity, specific lipase productivity, volumetric productivity, and total biomass activity.

Assay Protocol	Extracellular Hydrolytic Activity	Specific Productivity (U/g·h)	Volumetric Productivity (U/L·h)	Total Biomass Activity (U)
Soy Protein (50%) + No Oil	4.22	1.01	3.02	21.77
Soy Protein (50%) + Soy Oil	18.42	2.27	23.24	167.34
Soy Protein (100%) + No Oil	8.63	1.55	11.94	85.94
Soy Protein (100%) + Soy Oil	9.56	1.92	15.12	108.89

**Table 3 foods-14-00990-t003:** Initial reaction speeds in the different heating systems.

Heating System	Hydrolysis (%)	*v* (mmol/L·h)
Convencional	31.03 ± 0.40	27.92
Ultrasound	43.36 ± 0.11	70.26

## Data Availability

The original contributions presented in this study are included in the article. Further inquiries can be directed to the corresponding authors.

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
