# Peer review of "Sustainable Alternative Media for the Production of Lipolytic Cells and Fatty Acid Concentrates: Integration of the Enzyme and Food Industries"

_foods, 2025, doi:10.3390/foods14060990_

Round 1
Reviewer 1 Report
Comments and Suggestions for Authors
The study effectively highlights the potential of agro-industrial by-products, particularly soybean protein processing water, as a resource for sustainable enzyme production. However, it would be beneficial to include a comparative discussion on the environmental impact reduction achieved compared to conventional nitrogen sources.
The study presents clear findings on the optimal culture medium composition, with 50% textured soy protein yielding the highest hydrolytic activity. However, additional insights into why this specific composition was the most effective (e.g., amino acid profile, enhanced microbial growth) would strengthen the discussion.
It is noted that experiment does not mention microbial strains used for whole-cell lipase production, fermentation conditions, or whether the ultrasound method was optimized. Clarifying these aspects in the full article would strengthen the reproducibility and applicability of the findings.
Comments on the Quality of English LanguageThe quality of English in the manuscript is generally clear and professional, but there are some areas where minor improvements could enhance readability and precision.
Author Response
Please find the comments to Reviewer #1 attached.

Reviewer 2 Report
Comments and Suggestions for Authors
The paper examines sustainable lipase production from agro-industrial waste and its application in fatty acid production. It compares traditional hydrolysis methods with ultrasound-assisted hydrolysis. The findings are valuable and may enhance enzyme technology and sustainable bioprocessing.
I have reviewed your paper with keen interest and have provided detailed comments below to help further enhance your work.
• Please provide a more detailed description of "soybean oil residue." It is crucial to specify the source of this residue, indicating whether it originates from a restaurant kitchen or an industrial frying process. Please also clarify if it was gathered from a particular cooking method, such as deep-frying or pan-frying.
• Was the soybean oil residue filtered or pre-treated in any manner before being used in the culture medium and hydrolysis reactions? If so, could you please describe the pre-treatment process?
• It would be advantageous to include details regarding the composition (fatty acid profile, free fatty acid content, moisture content) of the soybean oil residue.
• In Figures 1B and 2, the Y-axis label is currently "Biomass (g/L)". Kindly change the label to "Biomass Concentration (g/L)" for greater precision and clarity.
• In section 3.2, when discussing the positive effect of soybean oil residue and the potential role of unsaturated fatty acids such as oleic and linoleic acid, consider reinforcing this point by citing specific studies that have explored the effect of these fatty acids on lipase production in Mucor circillenoides or closely related fungal species.
• The conclusions lack specificity and do not present the study's key results.
Comments on the Quality of English Language
There are style and grammar errors in the manuscript.
Author Response
Please find the comments to Reviewer #2 attached.
